# LeishTec vaccination disrupts vertical transmission of *Leishmania infantum*

**Diogo G. Valadares**[1], Eric Kontowicz[2¤a], Serena Tang[3], Angela Toepp[3¤b], Adam Lima[3¤c], Mandy Larson[4¤d], Tara Grinnage-Pulley[4¤e], Breanna Scorza[3], Danielle Pessoa-Pereira[3], Jacob Oleson[5], Christine Petersen[3¤f]*

1 Department of Internal Medicine, University of Iowa, Carver College of Medicine, Iowa City, Iowa, United States of America, 2 Department of Emergency Medicine, University of Iowa Hospital and Clinics, Iowa city, Iowa, United States of America, 3 University of Iowa, College of Public Health, Iowa city, Iowa, United States of America, 4 Department of Epidemiology, University of Iowa, College of Public Health, Iowa city, Iowa, United States of America, 5 Department of Biostatistics, University of Iowa, College of Public Health, Iowa city, Iowa, United States of America

¤a Current address: Department of Neurology, Iowa city, Iowa, USA.
¤b Current address: Sentara Healthcare, Norfolk, Virginia, USA.
¤c Current address: Universidade Estadual do Ceara´, Fortaleza, Ceara´, Brazil.
¤d Current address: Center for Veterinary Biologics, USDA-APHIS, Ames, USA.
¤e Current address: National Cancer Institute in Frederick, Frederick, Maryland, USA.
¤f Current address: Ohio State University, College of Veterinary Medicine, Columbus, Ohio, USA
* Diogo.Valadares@cshs.org

## Abstract

Zoonotic canine leishmaniosis, caused by *Leishmania infantum*, is a fatal disease worldwide in both humans and the reservoir host, dogs. The primary route of transmission is via sand fly bite. Vertical, transplacental, transmission of *L. infantum* to offspring has been shown to be critical for maintenance of infection in both endemic and non-endemic areas. In the United States, canine leishmaniosis (CanL) is enzootic within hunting dog populations. Previous work with US hunting dogs found that transplacental transmission of *L. infantum* occurs frequently with high infectivity. Dogs born to CanL infected mothers were almost fourteen times more likely to become positive for *L. infantum* over their lifetime. Globally, public health agencies control CanL through canine and human case detection and treatment, and in some cases dog culling and reducing vector populations. There is no specific strategy to control vertical transmission of CanL. A previous randomized field trial in US hunting dogs found that a *Leishmania* A2 protein, saponin-adjuvanted, vaccine (LeishTec) used as an immunotherapy, significantly reduced the risk of progression to clinically overt *leishmaniasis* by 30% in asymptomatic dogs. It is unknown whether maternal vaccination could inhibit infection risk in her offspring. We hypothesized that dogs born to infected and vaccinated dams would be less likely to test diagnostically positive via *L. infantum* specific kqPCR or serology compared to dogs born to infected unvaccinated mothers. A population of dogs born to *L. infantum* infected dams were evaluated to assess LeishTec vaccination to prevent transmission to offspring. Dogs born to

**Data availability statement:** All relevant data are in the manuscript and its Supporting Information files.

**Funding:** This work was supported by Morris Animal Foundation (C16CA-517 to CP). The funders had no role in the study design, data collection and analysis, decision to publish, or preparation of the manuscript.

**Competing interests:** The authors have declared that no competing of interests exist.

unvaccinated, *L. infantum* infected, dams had higher mortality (12.50% vs 0.00%), higher likelihood of clinical disease (94.12% vs 59.00%) and were more likely to be diagnostically positive for CanL (22.22% vs 4.55%). Vaccination of dams already infected prior to pregnancy greatly reduced the risk of transplacental transmission of *L. infantum*. Incorporating vertical transmission prevention as a public health intervention in countries where *Leishmania* is endemic could aid in infection control.

## Author summary

*Leishmania*sis, a parasitic disease, is a significant health concern for both humans and dogs worldwide, often transmitted through the bite of an infected vector called sand fly. While the disease is typically managed through case detection, treatment, and vector control, maternal transmission to offspring remains a critical factor in maintaining infection rates. In the United States, hunting dogs are particularly susceptible to canine leishmaniosis (CanL), with previous studies showing high rates of transplacental transmission from infected mothers to their puppies. The present study investigated the possible benefit of a *Leishmania* A2 protein vaccine, LeishTec, in disturbing maternal transmission of the parasite to offspring. Results showed that puppies born to vaccinated mothers had significantly lower mortality rates, reduced likelihood of developing clinical disease, more robust immunity against parasites and decreased rates of testing positive for CanL compared to puppies born to unvaccinated mothers. These findings suggest that vaccinating infected mothers before pregnancy can effectively inhibit vertical transmission of *L. infantum* in dogs. This research highlights the potential for vertical transmission prevention as new strategy for controlling CanL and reducing the burden of *leishmaniasis* in endemic areas. Larger and more effectively powered studies, focused on disrupting or preventing vertical transmission, are necessary to gain a clearer understanding of this phenomenon and assess its potential for integration into public health strategies.

## Introduction

Zoonotic canine leishmaniosis (CanL), caused by the obligate intracellular parasite *Leishmania infantum*, is a zoonotic disease of humans and dogs worldwide [1–6]. CanL is transmitted primarily when a phlebotomine sand fly takes a blood meal from human or other mammalian hosts [7,8]. Vector borne transmission occurs in countries where the disease is considered both endemic and enzootic within human and animal hosts respectively. Endemic areas of CanL include Asia, Africa, the Mediterranean basin, South and Central Americas [9,10]. Dogs are the main animal reservoir for CanL in many of these endemic regions and play a critical role in both the ecology and control of this disease [4]. It has been shown that infection of naïve animals, as

detected through seroprevalence, is a strong predictor for the emergence of human infection [11]. Dog ownership can be a risk factor for CanL [12–14].

In addition to sand fly transmission of parasites, there has been evidence of vertical, transplacental, transmission of parasites from mother to offspring during gestation [15]. In Brazil, an endemic country for CanL, some of the first reports of vertical transmission of *L. infantum* from dam (female, pregnant dog) to in utero puppies were presented [16,17]. Using polymerase chain reaction (PCR) and immunohistochemistry, parasites were found in stillborn puppies' liver and spleen samples. These puppies were born from a naturally infected dam with *L. infantum* [18]. In the United States, a country where CanL is an emerging problem, similar reports were found [17]. In a novel report, *L. infantum* was identified via *L. infantum*-kinetoplast specific quantitative PCR (kqPCR) in 8-day old puppies born to a naturally infected seropositive and kqPCR positive dam with no travel outside of the US [19]. Studies in a cohort of hunting dogs showed that vertical transmission maintains CanL incidence and prevalence in this population [4,20]. Through Bayesian compartmental models, it was found that the basic reproductive number (R0) for *L. infantum* in puppies born to infected dams was 3.37 with a 95.00% credible interval of (2.54, 4.04). This study also found that puppies born to CanL infected dams were more likely to progress to infection over their life course compared to those born to non-infected dams [21]. Despite a growing volume of literature on vertical transmission of *L. infantum* from dam to offspring, there is a knowledge gap regarding means to control vertical transmission in this critical reservoir population.

Between 2015 and 2017 a randomized, controlled, double-blinded field trial was performed with a cohort of hunting dogs to assess effectiveness of a *Leishmania* vaccine as an immunotherapy [22]. This study found that this immunotherapeutic vaccine reduced the risk of progression to clinically overt *leishmania* by 30% in asymptomatic dogs (RR: 1.33 95% C.I. 1.009-1.789) [22]. With prior research suggesting a transplacental transmission [19] and dam CanL infection status being a predictor of CanL in offspring [4], using dams vaccinated as part of this trial that became pregnant during the recruitment period, we hypothesized that dogs born to vaccinated, CanL-infected dams would be less likely to test positive via kqPCR or serology during post-field trial surveillance compared to dogs born to unvaccinated CanL infected dams. We further hypothesized that dogs born to placebo-vaccinated, CanL infected dams would test positive at a younger age compared to those dogs born to vaccinated, infected mothers.

This study examines offspring of two groups of *L. infantum* infected dams from hunting dog kennels within Midwest - USA, where one group of dams received LeishTec vaccine as immunotherapy [22] while the other did not. Analysis of offspring CanL status was performed from 2017 to 2020 via kqPCR and serology of dogs born to dams vaccinated during the original trial evaluating the association between dam's vaccination status and the probability of offspring testing positive for *L. infantum*.

## Methods

### Ethics Statement

All dogs were enrolled with signed informed consent and followed the protocol approved by the University of Iowa Institutional Animal Care and Use Committee (IACUC), an AAALAC (Association for the Assessment and Accreditation of Laboratory Animal Care International) accredited institution.

### Study design

A retrospective cohort study was performed on puppies that were born to dams enrolled in a block-randomized, double-blinded, placebo-controlled, immunotherapy trial for *L. infantum* [22]. A subset of dams (pregnant females) that were diagnostically positive for *L. infantum* infection via kqPCR, Dual-Path Platform Canine Visceral *Leishmaniasis* (DPP CanL) or soluble *Leishmania* antigen ELISA were excluded from Toepp et al. 2018 [22] due pregnancy and were included in the cohorts of the present study. Surveillance reports from 2017 onward were collected for puppies born to these dams (**Fig 1**).

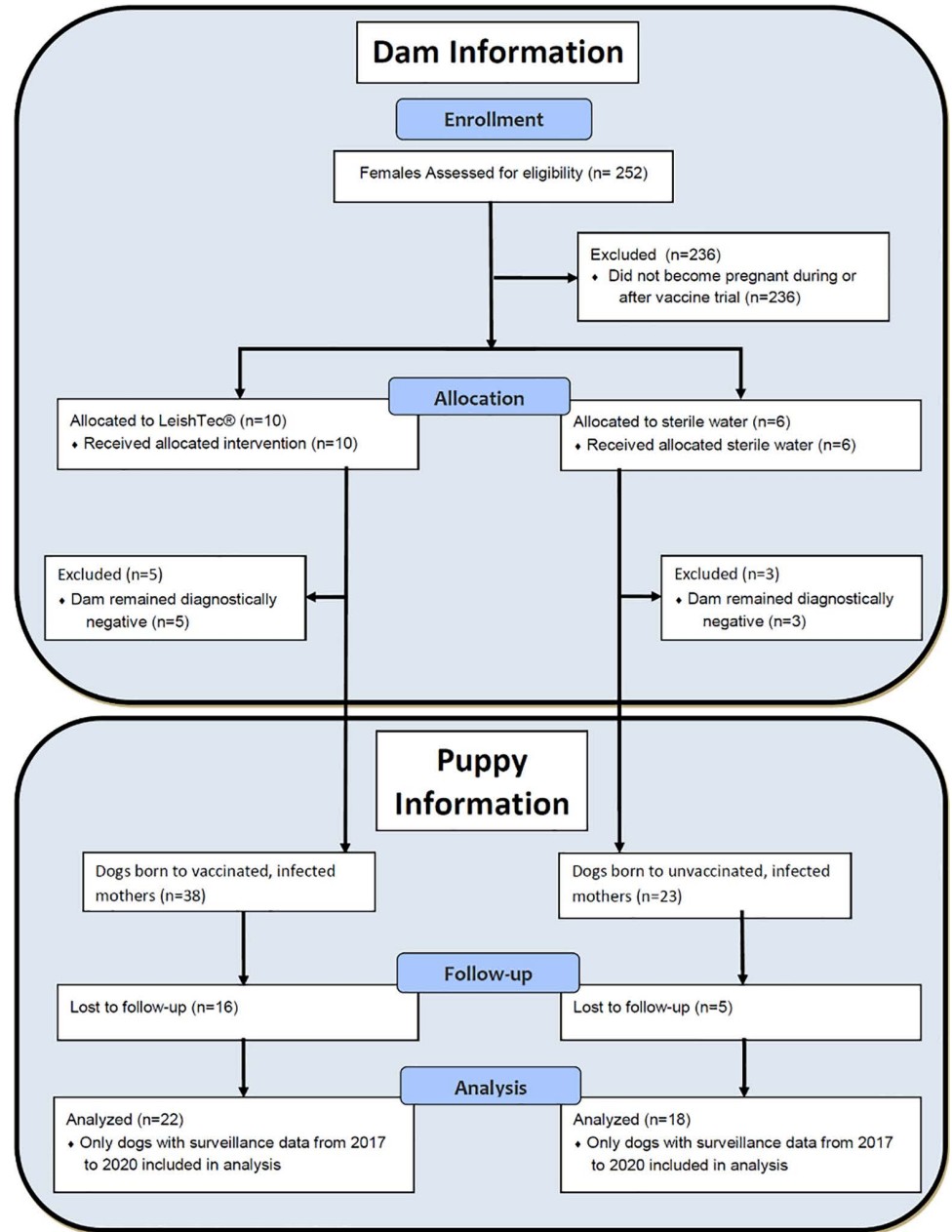

**Fig 1. CONSORT diagram for dam and puppy study inclusion.**

## Vaccine protocol

LeishTec, a recombinant A2-targeted, Quil A adjuvanted vaccine (Lot 042/15, Ceva Animal Health, Brazil) [23] was imported into the US from Brazil (permit no: VB-150792BRA). The clinical team obtained permission from the state veterinarian in each participating state to administer the experimental vaccine to animals within the state's borders, with the vaccination period for the trial beginning in February of 2016. For therapeutic vaccination strategy, female dogs received either three 1 mL subcutaneous injections (3 ml total/dose) in the left flank of LeishTec vaccine or vaccine diluent (sterile

water). To be considered fully vaccinated, female dogs were vaccinated three times at fourteen-day intervals with a 22 gauge, 1″ needle (Days 0, 14, and 28). Dogs were observed for one hour after administration of each full dose to monitor possible adverse events [24]. Vaccine and samples were kept at 4°C while in the field. Upon arrival at the laboratory samples were immediately processed and stored at −20°C (sera) or −80°C (whole blood). All blood and serum samples were obtained and stored with unique barcode identifiers.

## Experimental groups

The offspring of vaccinated dams group included puppies (N = 22) born to pregnant, CanL infected, dams (N = 5) that received complete immunotherapy (all 3 doses of LeishTec). Eighteen puppies born to three placebo-vaccinated, CanL infected dams composed the unvaccinated group. Female dogs with progressed clinical disease were excluded from the original vaccine study. This selection was performed based on laboratory records and number of immunotherapeutic doses the dam received. Animals were monitored over the course of 3 years with a total of 15 visits, occurring in intervals of 2–3 months, for physical examination for clinical signs of CanL; such as alopecia, dermatitis, conjunctivitis, epistaxis, cachexia, hepato- and/or splenomegaly, and lymphadenopathy; but also for diagnostic parameters including SLA-ELISA and parasite kDNA investigation in blood samples by kqPCR. A small number of dogs (20%, n = 8) from the litters also had complete blood count (CBC) and chemistry panel analysis allowing assignment of a LeishVet score, based on Solano-Gallego et. al 2011 [25].

## Parasite DNA isolation and kqPCR

QIAamp DNA Blood Mini Kit (QIAGEN, Valencia, CA) was used for DNA isolation per manufacturer's specifications for 200uL blood. Quality and quantity of isolated DNA was assessed by readings at A260 and A280 using a NanoDrop 2000 (Thermo Scientific, Waltham, MA). Isolated DNA (neat and 10-fold dilution) was analyzed in duplicate via RT-PCR in a 96-well plate format via Super Master mix with ROX (Quanta Biosciences, Gaithersburg, MD). Each kqPCR plate contained negative control nuclease-free water and samples of whole blood-extracted DNA from negative dogs. Positive control samples of $10^6$ *Leishmania* parasites spiked into healthy canine blood and subsequent DNA extraction were tested at full-strength, 1:10, and 1:20 dilutions for each plate. Ribosomal primer sequences: F 5'-AAGTGCTTTCCCATCGCAACT, R 5' CGCACTAAACCCCTCCAA (Invitrogen, Life Technologies, Grand Island, NY), probe: 5' 6FAM-CGGTTCGGTGTGTGG CGCC-MGBNFQ (Applied Biosystems, Life Technologies, Grand Island, NY). Primers and probes were used at a concentration of 10nM. The assay was performed on an ABI 7000 system machine. Thermocycle profile: 95°C for 2 min, 95°C for 1 min, and 50 cycles of 95°C for 15 seconds, 60°C for 1 min. kqPCR results were analyzed using ABI 7000 System SDS Software (Applied Biosystems, Life Technologies, Grand Island, NY).

## Dual Path Platform Canine Visceral *Leishmaniasis* (DPP CanL) assay

DPP CanL test detects *Leishmania*-specific antibodies using a recombinant *Leishmania* antigen, rK28, and colloidal gold particles coupled to protein A. A single positive control line was confirmed at 4 minutes or less. All positive or questionable samples were confirmed using the DPP Micro Reader (Chembios microreader system) which detects the intensity of the control and test lines within the test window and reports out an intensity value and result (negative or positive).

## Soluble *Leishmania* Antigen Enzyme-linked Immunosorbent (SLA-ELISA) assay

Peripheral blood draw was collected from dogs and serum samples were prepared, stored at -20°C and tested for antibodies against *L. infantum* SLA by ELISA method. 96-well plates (Costar-Fisher Scientific) were coated with SLA *L. infantum* (2 µg/well) overnight at 4°C. Next, plates were washed twice with PBS Tween 20 0.05%, and the wells were blocked with a PBS containing 1% BSA and 0.05% Tween 20, for 2 h at 37°C. The plates were then washed twice under the same

conditions, and serum samples (1:500 diluted) were added, in duplicate, for 2 h at room temperature (RT). After, plates were washed five times and incubated with Peroxidase-conjugated AffiniPure anti-dog IgG (H + L) (Jackson ImmunoResearch) was added (1:20,000 diluted), for 1 h at RT. Next, plates were again washed five times, color developed using TMB (BD OptEIA) for 15 min in the dark, and the reaction stopped with addition of 100 μL 2M sulfuric acid. Optical densities were read at 450 nm in an ELISA microplate spectrophotometer (VersaMax, Molecular Devices).

### Flow cytometry

Evaluation of T cell immunity was performed by isolating peripheral blood mononuclear cells (PBMCs) from canine whole blood samples. Cells were stimulated with total *leishmania* antigen (TLA) (10ug/mL), for 7 days, to assess specific cytokine production against *Leishmania* parasites. For intracellular cytokine detection, cells were fixed and permeabilized using fixation buffer (BioLegend Inc). For intracellular staining of CD4 + cells, zenon conjugation kits were used to label anti-canine IFNγ-R-PE (5 μg/mL, R&D Systems) and anti-canine IL-10-AF488 (5 μg/mL, R&D Systems) antibodies (ThermoFisher Z25255 and Z25002, respectively) according to manufacturer instructions. Flow cytometric analysis was conducted using a Becton Dickenson LSR II flow cytometer and data were analyzed using FlowJo software. Data were analyzed to determine the frequency and functionality of CD4 + cell populations, focusing on cytokine production profiles.

### Outcome

The study objective was to determine the effect of CanL immunotherapy at reducing the odds of vertical transmission of *L. infantum* between infected dams and their offspring. Our hypothesis is that a productive immune response produced via vaccine immunotherapy of CanL infected pregnant dogs disrupts vertical transmission of *L. infantum*. To assess this relationship, puppies were considered CanL positive if they ever tested positive via kqPCR, DPP CanL or SLA-ELISA. Whole blood and serum samples were collected from puppies at 2–3-month intervals in a total of 15 visits and used in diagnostic testing.

### Data management

All blood and serum samples were obtained and stored with unique identifiers. All dog names and matching identifiers were securely stored in Microsoft Excel spreadsheets only accessible to designated research team members to maintain unbiased physical examination and diagnostic testing. The main data used for generating manuscript figures is available as supporting information (S1 Data).

### Statistical analysis

Toepp et al. 2018 [22] showed that dogs that did not receive the immune therapy during the clinical trial had a 33.00% increase in risk of progressing to diagnostically positive status. Prior studies of our group had shown that 10.00% of all dogs, regardless of vaccination status, would progress from negative to diagnostically positive for CanL. Based on these values, sample size calculation revealed the need of 27 puppies per group to achieve 80.00% power for diagnostic purposes at the 0.05 alpha level. For this study, due to recruitment limitation, we analyzed 18 puppies born from infected unvaccinated dams and 22 from infected vaccinated dams.

A Fisher's exact test was used to evaluate the association between the dam vaccination status and their offspring became diagnostically positive to *L. infantum* at any of the surveillance visits following the initial vaccine trial. Additionally, a weighted chi-squared analysis was used to analyze the association between dam vaccination status and offspring's diagnostic positivity to *L. infantum*. Given the small sample size of recruited puppies for this analysis and the important impact of age on the risk of testing positive for *L. infantum* in our cohort [26–28], we implemented a weighted analysis to account for the age distribution of the sampling population. Weight factors (Table 1) were computed based on the ratio of

Diseases

**Table 1. Weighted factors calculated based on year of birth distributions of sample and entire puppy population.**

| Year of Birth | Sample (n = 40) | Population (n = 117) | Age-based weight factor |
|---|---|---|---|
| 2016 (%) | 45.00 | 44.44 | 0.988 |
| 2017 (%) | 42.50 | 18.80 | 0.442 |
| 2018 (%) | 12.50 | 7.69 | 0.615 |

offspring born to dams enrolled in the vaccine trial (n = 117) to all offspring included in this study. These weighted factors were calculated for dogs born in the years 2016, 2017, and 2018, corresponding to the birth years of dogs in this study.

The number of clinical signs between groups were compared using a two-sample T-test. Statistical significance for all statistical tests were defined as p-values below 0.05. A bivariate time to event analysis was performed to assess the effect of dam's vaccination status with time to offspring's first positive diagnostic test using a Kaplan-Meier curve and Cox proportional-hazard model. All analyses were performed in Graph Prism (version 8 - La Jolla, CA) or R statistical software (Version 3.6.2; R Foundation of Statistical Computing, Vienna, Austria).

## Results

To evaluate a relationship between maternal vaccination, as immunotherapy, and its impact on offspring CanL positivity, we compared CanL outcomes from puppies born to vaccinated, CanL infected dams and placebo-vaccinated, CanL infected dams. The study design was a retrospective, nested, vaccine clinical trial using dam vaccine exposure data from a previous larger blinded, placeboed, vaccine clinical and pup outcome data from multiple different surveillance studies performed by the Petersen lab between 2016 and 2022. Between December of 2015 and February 2016, 557 dogs were selected and randomized for vaccine as part of a LeishTec immunotherapy trial [22]. Among two hundred fifty-three (253) female dogs, one hundred and thirty-one (131) of these females received the vaccine and the remaining one hundred and twenty-one (122) females received placebo. A total of 236 female dogs were excluded from this study because they did not become pregnant during or after the vaccine trial or were clinically symptomatic for CanL. Data from sixteen females was used in this study, with exclusion of 8 dogs. The remaining cohort of females consisted of eight CanL positive dogs as assessed via kqPCR, DPP CanL or SLA-ELISA between January and November of 2016. Five of the selected females received the vaccine and three received placebo, giving birth to 61 puppies either during or following the vaccine trial (**Fig 1**). Of those 61 puppies, outcome data was available for a total of 22 puppies born to vaccinated, CanL infected dams and 18 puppies born to unvaccinated, CanL infected dams were used in this study, total n = 40, 65.6% of offspring. The remaining 21 (34.4%) puppies were excluded due to loss to follow up, typically being traded to outside hunting kennels or dogs being unsuccessful hunters and adopted out to the public.

The demography of puppies included in this retrospective study had unequal, but relatively close distribution of age and sex for each group (**Table 2**). Unvaccinated dams were noted to be pregnant earlier in the vaccine trial and thus gave birth to puppies before vaccinated dams. We found unvaccinated dams gave birth to puppies approximately 0.89 years before vaccinated dams. All dogs included in this analysis were from Midwestern United States.

To evaluate maternal vaccination status (LeishTec immunotherapy) and its impact on clinical manifestation of CanL in offspring, we evaluated mortality, manifestation of clinical signs typical of CanL and *L. infantum* infection markers such as serology against parasite antigens (by DPP or ELISA) and parasitemia by kqPCR (**Fig 2**). Puppies born to vaccinated, CanL infected dams were found to have 100% survival during the study period. Puppies born to placebo vaccinated, CanL infected dams had a mortality rate of 12.50% (**Fig 2A**). Deaths occurred mainly during the first year of age with one loss due kidney failure at 3–4 years of age. A higher percentage (94.12%) of dogs born to placebo vaccinated, CanL infected dams manifested clinical signs of CanL over the study period compared to dogs (59.00%) born to vaccinated CanL infected dams (**Fig 2B**). Puppies born to vaccinated, CanL infected dams had an average of 1.50 clinical signs. Similarly,

**Table 2. Demographics of animals enrolled in the study. Vaccination regime was considered complete with 3 doses of LeishTec.**

| Variable | Dam Not Vaccinated group | Dam Vaccinated group |
|---|---|---|
| Dams (number) | 3 | 5 |
| Age of dams at the study enrollment (years – mean +/- sd) | 4.67 ± 1.5 | 5 ± 1.29 |
| Number of LeishTec® vaccine doses given | 0 | 3 |
| Litter size (mean +/- sd) | 7.67 ± 1.53 | 7.6 ± 2.07 |
| Number of puppies enrolled in study | 18 | 22 |
| Sex, % male | 50.00 | 63.63 |
| Age of puppies in years at end of study, mean +/- sd | 3.66 ± 0.48 | 2.77 ± 0.42 |

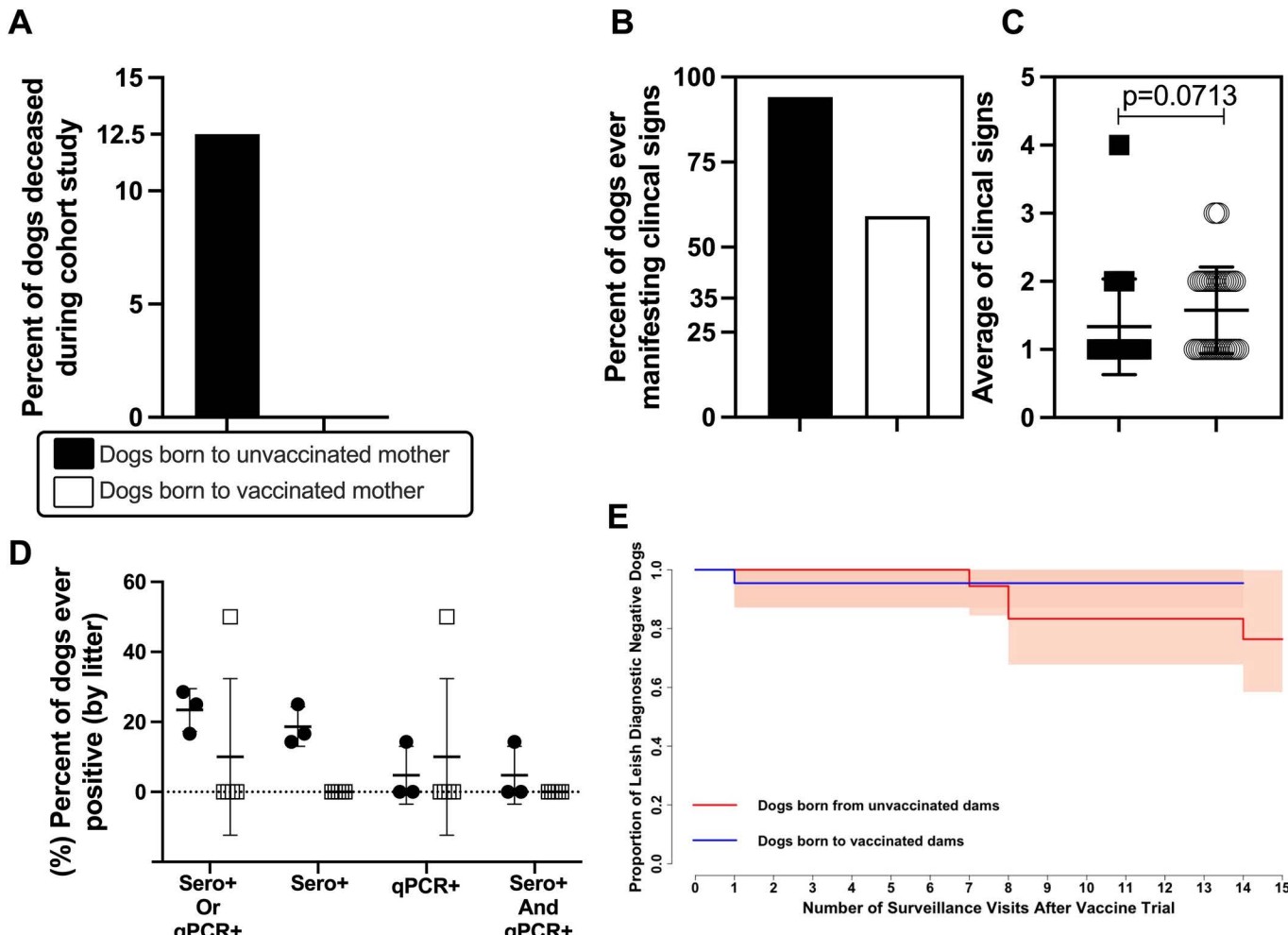

**Fig 2. Evaluation of mortality, clinical signs and canine leishmaniosis diagnosis in puppies born to infected and placebo-vaccinated or infected and vaccinated dams.** Dogs born to placebo-vaccinated or vaccinated dams were counted and graphed based on mortality **(A)**, manifestation of the number of CanL associated clinical signs **(B, C)**, CanL diagnosis by litter, assessed by kqPCR, DPP or ELISA **(D)** and loss of negative diagnosis for CanL **(E)** Kaplan-Meier curve for dogs born to vaccinated (blue line) and unvaccinated (red line) infected dams. Graphs and analysis were performed in Graph Prism (version 8) and R (version 3.6.2).

puppies born to placebo vaccinated, CanL infected dams had an average of 1.30 clinical signs (**Fig 2C**). Just one dog born from the vaccinated group was positively diagnosed for CanL. This subject had a transient kqPCR positive diagnosis that was detected only once but remained serologically negative during the study period. Meanwhile, 22.20% of puppies born to placebo vaccinated CanL infected dams had positive diagnosis for CanL (**Fig 2D**). Among positively diagnosed puppies, born to infected unvaccinated dams, 16.67% were DPP or SLA-ELISA positive, 11.11% kqPCR and 5.50% (1 dog) was positive on both kqPCR and serology (DPP or SLA-ELISA) (**Fig 2D**). Most dogs born to vaccinated, CanL infected dams (95.45%) demonstrated no diagnostic positivity for CanL during the study period.

Fisher's exact test comparing mother's vaccination status to puppies' diagnostic outcome demonstrated that offspring from vaccinated mothers had reduced odds (uOR = 0.17, 95%CI: 0.003-2.00) of testing positive for CanL during the follow-up period. A weighted chi-squared analysis (dams vaccination status and positive diagnosis of respective litter) found similar results (Chisq = 2.33, p = 0.13). Although reduced odds of testing positive for CanL were obtained in the cohort from vaccinated dams comparing with pups from unvaccinated dams, such reduction was not significant probably due sample size limitations and reduced power. Larger cohorts would potentially indicate significant results. All dogs that were diagnostically positive were 3–4 years of age, and none of the dogs younger than three were found to be diagnostically positive. A Kaplan-Meier curve (**Fig 2E**) shows that with the exception of one dog, all offspring born to vaccinated mothers remained diagnostically negative at each surveillance visit following the vaccine trial. A Cox proportional-hazards model suggests that offspring born to infected vaccinated dams had a lower hazard (HR = 0.21 [0.02 – 1.89]) of testing diagnostically positive compared to offspring born to infected placebo vaccinated dams.

*Leishmania*sis is an immunosuppressive disease, where CD4 secreting IFN-γ cells have been shown to be required for parasite control, and IL-10 levels are related to parasite persistence [29]. Dogs born to vaccinated dams showed a gradual increase on the proportion of CD4 IFN-γ/CD4 IL-10 secreting cells, while animals born to unvaccinated dams failed demonstrate an increase in this ratio (**Fig 3A**). On other hand, dogs born to unvaccinated dams showed a higher Leishvet clinical score, ranging from 2 to 4, which is associated with worse disease manifestations and more clinical complications, while dogs born to vaccinated dams showed a score raging from 0 to 2 (**Fig 3B**). Immunotherapy before or during a dam's pregnancy triggered an efficient immune response on the resultant litter whelped in the year to follow, increasing key immune factors responsible for parasite control, and may respond partially for the better outcome of dogs born to vaccinated dams.

## Discussion

As shown in previous studies, CanL is maintained through transplacental transmission in a cohort of United States hunting dogs [19,20,30]. The use of a *Leishmania* vaccine as an immunotherapy reduces the risk of disease progression and mortality in these dogs [22]. Blocking transmission of this parasite during the gestation is critical to prevent the spread of CanL in dogs. This current work suggests that dogs born to infected dams which received a vaccine, as immunotherapy, were less likely to be diagnostically positive compared to dogs born to dams that received a placebo vaccine. It was further shown dogs born to vaccinated dams had a longer time to diagnostic positivity compared to those dogs born to unvaccinated dams. These findings suggest that a vaccine as an immunotherapy before or during a dam's pregnancy can reduce transplacental transmission of *L. infantum* from infected mothers to gestating puppies.

In our cohort of dogs, only one of the dogs born to vaccinated, CanL infected dams was diagnostically positive (4.55%) in the three years of post-trial surveillance. Conversely, four of eighteen (22.22%) dogs born to placebo vaccinated mothers were diagnostically positive. Our findings suggest that LeishTec immunotherapy of female dogs can likely significantly/drastically decrease transplacental transmission of parasites from mother to offspring. Although sandfly populations within the USA are spreading from southern to more northern regions of the country, CanL remains enzootic without evidence of vector borne transmission [31,32]. This highlights the importance of previous studies demonstrating that transplacental transmission is epidemiologically maintaining canine infection and disease inside US borders, which is also likely to

PLOS Neglected Tropical Diseases

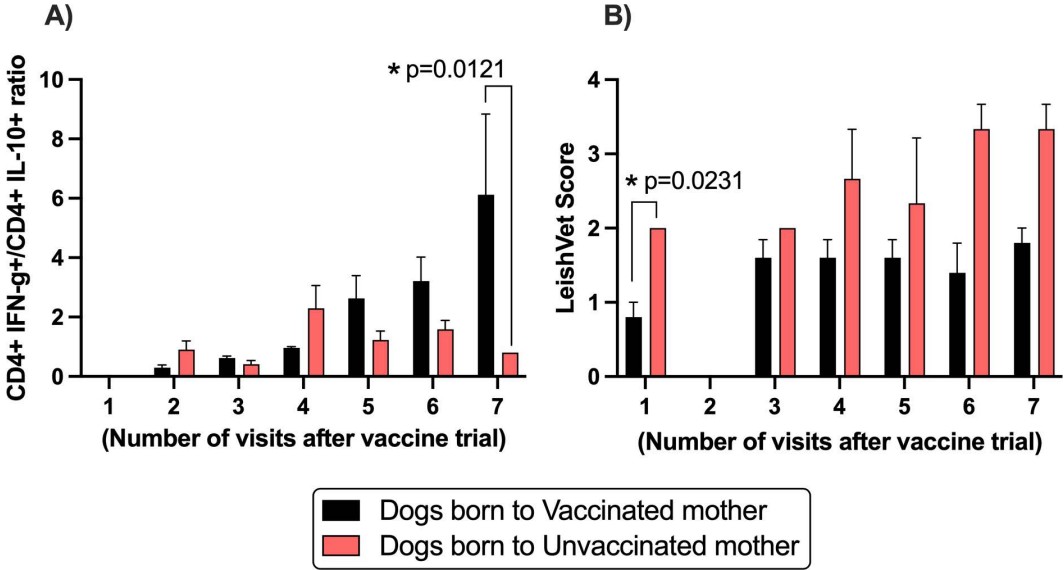

**Fig 3. Evaluation of adaptive response and Leishvet clinical in puppies born to infected and placebo-vaccinated or infected and vaccinated dams.** Adaptive responses measured by ratio of CD4 IFN-g by CD4 IL-10 secreting cells was done, comparing dogs born to placebo-vaccinated or vaccinated dams **(A)**. Clinical manifestation, based on LeishVet system, of the number of CanL, was also performed on both cohorts. Data was acquired every 3 months from 2019 – 2021. Graphs and analysis were performed in Graph Prism (version 8) using 2way ANOVA. Dogs born to vaccinated mothers: 5, dogs born to unvaccinated mothers: 3.

occur in other regions where sandfly populations are small or inconsistent [19]. It has been previously estimated that one infected dam can give birth to four infected puppies (R0 = 4) [4]. Our findings suggest that control of this enzootic disease could be achieved by adding specific measures to address vertical transmission, such as vaccination of infected females or sterilization strategies, compared to dog culling which has been shown to be ineffective [33]. Our study also unveil the possibly a benefit of vaccinating all at risk animals (uninfected or infected) to reduce the chance of vertical transmission that may occur later during dogs pregnancy.

The key limitation of this study is the small sample size and small number of events. These two limitations minimized analysis options. However, this is a unique cohort addressing strategies that can influence vertical transmission of *Leishmania* parasites in canine infection. It is likely these observed differences would remain and become more significant with a larger sample, while the odds and hazard ratios would likely be more protective. Not all dogs were the same age in this study, and a traditional multivariate analysis was not able to be performed to allow for adjustment of age differences in these populations. However, results from a weighted chi square test suggested that observed differences were not confounded by age. Reduction of diagnostic positivity in animals born to infected and vaccinated dams does not equate to complete blockade of vertical transmission, but more likely disruption and impairment of vertical transmission. There is still the possibility of offspring with low levels of infection more likely to cause subclinical disease. Unfortunate events including comorbid tick-borne disease and others could still lead to expansion of parasites and CanL [34]. Larger, better powered studies, aiming to disrupt or impede vertical transmission, are needed to better understand this phenomenon. Ideally, cohorts including outbred dogs from different endemic regions would provide important information regarding the efficacy and feasibility of such an innovative control strategy.

Reducing the impacts of *Leishmania* infection and its progression via immunotherapy has been shown to improve the quality of life of both animals and humans alike [35,36]. Previous work has shown that implementation of a *Leishmania* vaccination as an immunotherapy has direct positive effects on reducing disease progressions and mortality in

asymptomatic dogs [22,25,34]. Results of this study show that offspring born to vaccinated, infected dams are less likely to test diagnostically positive for *Leishmania*. It is imperative to highlight that the intervention of vaccination (immunotherapy) of mothers, had an impact on offspring CanL diagnosis without any direct intervention in the puppies. Immunotherapy (with LeishTec vaccine) of mothers followed by puppy vaccination could potentially protect offspring from ever manifesting disease. Further studies to assess this would provide valuable tools to progress towards elimination of zoonotic Visceral *Leishmaniasis*. Vaccine as an immunotherapy had a positive impact on adult dogs and showed significant disruption of *Leishmania* vertical transmission to offspring, providing health benefits to two generations of dogs.

## Supporting information

**S1 Data.**

(XLSX)

## Author contributions

**Conceptualization:** Diogo G Valadares, Eric Kontowicz, Christine Petersen.

**Data curation:** Diogo G Valadares.

**Formal analysis:** Eric Kontowicz.

**Investigation:** Serena Tang, Angela Toepp, Adam Lima, Mandy Larson, Tara Grinnage-Pulley, Breanna Scorza, Danielle Pessoa-Pereira.

**Resources:** Angela Toepp, Adam Lima, Danielle Pessoa-Pereira, Christine Petersen.

**Supervision:** Christine Petersen.

**Visualization:** Diogo G Valadares, Eric Kontowicz, Jacob Oleson, Christine Petersen.

**Writing – original draft:** Diogo G Valadares, Eric Kontowicz, Mandy Larson, Tara Grinnage-Pulley, Jacob Oleson, Christine Petersen.

**Writing – review & editing:** Diogo G Valadares, Eric Kontowicz, Christine Petersen.

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
