## [Decision Letter · Decision Letter 0]

PNTD-D-24-01331

LeishTec Ⓡ vaccination disrupts vertical transmission of Leishmania infantum

Dear Dr. Valadares,

Thank you for submitting your manuscript to PLOS Neglected Tropical Diseases. After careful consideration, we feel that it has merit but does not fully meet PLOS Neglected Tropical Diseases's publication criteria as it currently stands. Therefore, we invite you to submit a revised version of the manuscript that addresses the points raised during the review process.

Please submit your revised manuscript within 60 days Feb 04 2025 11:59PM. If you will need more time than this to complete your revisions, please reply to this message or contact the journal office at plosntds@plos.org. Please include the following items when submitting your revised manuscript:

We look forward to receiving your revised manuscript.

Kind regards,

Deborah Bittencourt Mothé Fraga, Ph.D.

Guest Editor

Guilherme Werneck

Section Editor

Shaden Kamhawi

co-Editor-in-Chief

Paul Brindley

co-Editor-in-Chief

**Additional Editor Comments:**

The study is highly significant as it investigates the use of a Leishmania vaccine in preventing the vertical transmission of L. infantum from infected mothers to their offspring. Based on my evaluation of the manuscript, I believe it requires major revisions before it can be accepted for publication.

Despite differing opinions among the reviewers (one recommending rejection, another suggesting major revisions, and a third suggesting minor revisions), I believe that undertaking major revisions is essential to improve the manuscript and make it suitable for publication.

As Reviewer 2 pointed out, PCR was performed on blood samples, which are not the most sensitive samples for this purpose. The authors should justify this choice and consider using more appropriate samples, such as bone marrow or spleen tissue. Additionally, it would be valuable to include an analysis of laboratory parameters, such as total protein and its fractions, creatinine levels, urinalysis, and UPC (urine protein-to-creatinine ratio).

As Reviewer 3 recommended, the manuscript should include an immunological analysis to explain how the vaccine blocks transmission to the puppies. Is this achieved through a humoral immune response, a cellular immune response, or both? A semi-quantitative immunological assessment of both the dams and the puppies should be conducted to address this question.

**Journal Requirements:**

At this stage, the following Authors/Authors require contributions: Diogo G Valadares, Erick Kontowicz, Serena Tang, Angela Toepp, Adam Lima, Mandy Larsen, Tara Grinnage-Pulley, and Jacob Oleson. Please ensure that the full contributions of each author are acknowledged in the "Add/Edit/Remove Authors" section of our submission form.

4) We do not publish any copyright or trademark symbols that usually accompany proprietary names, eg ©,  ®, or TM  (e.g. next to drug or reagent names). Therefore please remove all instances of trademark/copyright symbols throughout the text, including:

- ® on: Title, and pages 5, 7, 9, and 14.

5) Your manuscript is missing the following sections: Discussion.  Please ensure all required sections are present and in the correct order. Make sure section heading levels are clearly indicated in the manuscript text, and limit sub-sections to 3 heading levels. An outline of the required sections can be consulted in our submission guidelines here:

6) Please upload all main figures as separate Figure files in .tif or .eps format. For more information about how to convert and format your figure files please see our guidelines:

7) We note that your Data Availability Statement is currently as follows: "The data supporting the findings of this study are available upon request through the REDCap data management platform. REDCap (Research Electronic Data Capture) is a secure, web-based application designed to support data capture for research studies, providing validated and auditable data collection. Access to the data is restricted to ensure patient confidentiality and adherence to ethical guidelines.Researchers interested in accessing the dataset may contact the corresponding author to request access. To ensure proper data use, prospective users will be required to submit a brief description of their intended research use, agree to a data use agreement (DUA), and obtain necessary institutional approvals, such as Institutional Review Board (IRB) approval, if applicable.All data will be provided in a de-identified format to protect participant privacy. Specific variables and any associated metadata will be made available to qualified researchers following approval.". Please confirm at this time whether or not your submission contains all raw data required to replicate the results of your study. Authors must share the “minimal data set” for their submission. PLOS defines the minimal data set to consist of the data required to replicate all study findings reported in the article, as well as related metadata and methods (https://journals.plos.org/plosone/s/data-availability#loc-minimal-data-set-definition).

- The points extracted from images for analysis..

8) Please amend your detailed Financial Disclosure statement. This is published with the article. It must therefore be completed in full sentences and contain the exact wording you wish to be published. Please ensure that the funders and grant numbers match between the Financial Disclosure field and the Funding Information tab in your submission form. Note that the funders must be provided in the same order in both places as well. State the initials, alongside each funding source, of each author to receive each grant. For example: "This work was supported by the National Institutes of Health (####### to AM; ###### to CJ) and the National Science Foundation (###### to AM)." State what role the funders took in the study. If the funders had no role in your study, please state: "The funders had no role in study design, data collection and analysis, decision to publish, or preparation of the manuscript.".

**Reviewers' Comments:**

Reviewer's Responses to Questions

**Key Review Criteria Required for Acceptance?**

**Methods**

-Are the objectives of the study clearly articulated with a clear testable hypothesis stated?

-Is the study design appropriate to address the stated objectives?

-Is the population clearly described and appropriate for the hypothesis being tested?

-Is the sample size sufficient to ensure adequate power to address the hypothesis being tested?

-Were correct statistical analysis used to support conclusions?

-Are there concerns about ethical or regulatory requirements being met?

Reviewer #1: -Are the objectives of the study clearly articulated with a clear testable hypothesis stated? Yes

-Is the study design appropriate to address the stated objectives? Yes

-Is the population clearly described and appropriate for the hypothesis being tested? Yes

-Is the sample size sufficient to ensure adequate power to address the hypothesis being tested? No, but the authors acknowledge and address the sample size limitation.

-Were correct statistical analysis used to support conclusions? Yes

-Are there concerns about ethical or regulatory requirements being met? Yes

Reviewer #2: The objectives of the study is clearly articulated with a clear testable hypothesis stated but the study design is not appropriate for the hypothesis tested, as PCR was performed on a blood sample. This biological sample is not very sensitive for evaluating infection. It would be necessary to use a bone marrow or spleen sample. Furthermore, it would be important to analyze the laboratory dosages of total protein and fractions, creatine, urinalysis and UPC. Analyzing clinical signs alone is not sufficient to determine the presence of canine Leishmaniasis. Scientific studies demonstrate that there are patients without evident clinical signs with subclinical disease and moderate to high parasite load in the spleen and bone marrow.

The population clearly described and appropriate for the hypothesis being tested? Yes.

Is the sample size sufficient to ensure adequate power to address the hypothesis being tested? Yes.

Were correct statistical analysis used to support conclusions? Needs to be reviewed.

Are there concerns about ethical or regulatory requirements being met? Yes, there are.

Reviewer #3: -Are the objectives of the study clearly articulated with a clear testable hypothesis stated? Yes.

-Is the study design appropriate to address the stated objectives? No, the number of dams were not sufficient for a strong and comprehensive statistical analysis

-Is the population clearly described and appropriate for the hypothesis being tested? No, this is the major problem of the manuscript.

-Were correct statistical analysis used to support conclusions? Yes. However, with a limited force.

-Are there concerns about ethical or regulatory requirements being met? Yes

**Results**

-Does the analysis presented match the analysis plan?

-Are the results clearly and completely presented?

-Are the figures (Tables, Images) of sufficient quality for clarity?

Reviewer #1: -Does the analysis presented match the analysis plan? Yes

-Are the results clearly and completely presented? Yes

-Are the figures (Tables, Images) of sufficient quality for clarity? Yes

Reviewer #2: Does the analysis presented match the analysis plan? yes.

Are the results clearly and completely presented? No, needs to be reviewed.

Are the figures (Tables, Images) of sufficient quality for clarity? The flowchar and demographic data is well designed and clearly defines the analysis and losses between the groups. The graphs on clinical signs could be better detailed regarding the intensity of each clinical sign and which signs were most prevalent.The graphs on antibody quantification do not define the cutoff point nor does it define which symbol represents the offspring of each group. The survival curve of the dams shows that unvaccinated bitches had a longer survival rate. This information needs to be discussed at work.

Reviewer #3: -Does the analysis presented match the analysis plan? No, due to the limited number of dams included in the study.

-Are the results clearly and completely presented? No, several aspects of the results should be more addressed in the results, such as the humoral response.

-Are the figures (Tables, Images) of sufficient quality for clarity? Yes

**Conclusions**

-Are the conclusions supported by the data presented?

-Are the limitations of analysis clearly described?

-Do the authors discuss how these data can be helpful to advance our understanding of the topic under study?

-Is public health relevance addressed?

Reviewer #1: -Are the conclusions supported by the data presented? Yes

-Are the limitations of analysis clearly described? Yes

-Do the authors discuss how these data can be helpful to advance our understanding of the topic under study? Yes

-Is public health relevance addressed? Yes

Reviewer #2: Are the conclusions supported by the data presented? No.

Are the limitations of analysis clearly described? Yes.The results do not adequately assess the presence of infection or disease.

Spleen or bone marrow PCR should have been used to quantify the parasite load.

Total proteins and fractions, urinalysis and UPC should have been measured, among other biochemical analyzes that could evaluate canine leishmaniasis.

The authors conclude that the vaccination of dams allowed uninfected puppies, but they did not perform PCR on biological samples with a sufficient degree of sensitivity to define the parasite load.

The objective of the study is valid and its design is well defined, but the diagnostic tools were not adequate. It is a risk to public health, suggested that vaccination was able to block vertical transmission based on the results obtained in this work.

I suggest that the authors look into the possibility of collecting spleen or bone marrow samples for evaluation of these patients. And biochemical tests for analysis and staging of canine leishmaniasis according to the guide line by Solano-Gallego et al.

The discussion about the survival curve of dams should also be carried out.

Reviewer #3: -Are the conclusions supported by the data presented? No, due to the insufficient number of dams included in the study.

-Are the limitations of analysis clearly described? Yes.

-Do the authors discuss how these data can be helpful to advance our understanding of the topic under study? YEs, but with a limited emphasis.

-Is public health relevance addressed? Yes, but with a limited emphasis.

**Editorial and Data Presentation Modifications?**

Reviewer #1: Thank you for the opportunity to review. This paper details a study investigating whether vaccination of dams provides some protection against leishmaniosis in their offspring. The results are very interesting and promising, although the authors appropriately note the limitations of their sample size. The study seems well designed, and the paper is well-written overall. However, I think some minor revisions are necessary.

Reviewer #2: This article needs Major revision with modifications to its initial hypotheses or new tests on study patients.

Reviewer #3: (No Response)

**Summary and General Comments**

Reviewer #1: Thank you for the opportunity to review. This paper details a study investigating whether vaccination of dams provides some protection against leishmaniosis in their offspring. The results are very interesting and promising, although the authors appropriately note the limitations of their sample size. The study seems well designed, and the paper is well-written overall. However, I think some minor revisions are necessary, and did have a couple of clarifying questions for the author’s consideration:

- How were the dogs kept/housed? They are described as being recruited from hunting dog kennels in the Midwestern US. These were not research-purpose dogs? Noting that transmission via sand flies is not thought to occur in the U.S., but horizontal transmission is still possible – so wondering whether there were differences in how the dogs were housed/kept between the experimental and control groups that could have affected the outcome (e.g., were they housed singly, or in groups, etc.).

- Please provide the sensitivities and specificities of the diagnostic tests used.

- In the “results and conclusions” section, third paragraph, where the one dog born to a vaccinated dam and who had a transient positive kqPCR result is being discussed, there is this in the middle of the sentence: “(AGE?)”. I’m not sure what this means but think it would be important to know the age at which the dog’s positive result occurred.

Other edits:

- While the final manuscript will almost certainly undergo proofreading at the journal, I would encourage the authors to also do that before they submit their revision. For example, there are a couple of instances in the “results and conclusions” section where the word “dams” was misspelled. Additionally, the last sentence in the 6th paragraph in the “results and conclusions” section is a little garbled and could use some restructuring for clarity: “Our study also unveil the possibly a benefit of vaccinating all at risk animals…”

- Reference #15 and #19 are the same.

- Reference #20 and #27 also appear to be the same.

- Reference #4 and #26 are the same.

- Reference #28 and #22 are the same.

Reviewer #2: The idea of the article is excellent, but insensitive tools were used to test it. I hope that it will be possible to collect patient samples to perform PCR and staging of canine leishmaniasis.

It is necessary to use articles from the Leishvet group as scientific references.

Reviewer #3: This is an interesting study that shows the influence of vaccination in the transmission of Leishmania from infected dams to their puppies. However, several points of this study should be considered.

- The number of dams included in the study is very limited. In this way, several statistical analyses lack force and are not so strong. the results are very preliminary, and makes this manuscript more suitable as a short communication.

- Authors do not show what is important for the blockage of transmission for the puppies. Would it be the humoral response? the cellular immune response? A semi-quantitative analysis of the dams and the puppies would be interesting to aid to solve this problem.

- Additionally, it would bel also interesting a basic assay on the specific-production of IFN gamma by the dams and by the puppies, to sse if there is any correlation between the IFN gamma levels and the blockage of transmission.

- The manuscript could be send to a English grammar and structure revision service, since several parts of the manuscript are particularly difficult to understand.

PLOS authors have the option to publish the peer review history of their article (what does this mean? ). If published, this will include your full peer review and any attached files.

**Do you want your identity to be public for this peer review?** For information about this choice, including consent withdrawal, please see our Privacy Policy .

Reviewer #1: No

Reviewer #2: No

Reviewer #3: No

**Figure resubmission:**
---

## [Editor Report · Decision Letter 1]

Dear Dr Valadares,

We are pleased to inform you that your manuscript 'LeishTec vaccination disrupts vertical transmission of Leishmania infantum' has been provisionally accepted for publication in PLOS Neglected Tropical Diseases.

Best regards,

Deborah Bittencourt Mothé Fraga, Ph.D.

Guest Editor

Guilherme Werneck

Section Editor

Shaden Kamhawi

co-Editor-in-Chief

Paul Brindley

co-Editor-in-Chief

---

## [Editor Report · Acceptance letter]

Dear Dr Valadares,

We are delighted to inform you that your manuscript, "LeishTec vaccination disrupts vertical transmission of Leishmania infantum," has been formally accepted for publication in PLOS Neglected Tropical Diseases.

Best regards,

Shaden Kamhawi

co-Editor-in-Chief

Paul Brindley

co-Editor-in-Chief
